# Implicit NNs are Almost Equivalent to Not-so-deep Explicit NNs for High-dimensional Gaussian Mixtures

## Abstract

Implicit neural networks (NNs) have demonstrated remarkable success in various tasks; however, there is a lack of theoretical understanding of the connections and differences between implicit and explicit networks. In this paper, we employ random matrix theory (RMT) to analyze the eigenspectra of neural tangent kernels (NTKs) and conjugate kernels (CKs) for a broad range of implicit NNs, when the input data are drawn from a high-dimensional Gaussian mixture model. Surprisingly, the spectral behavior of Implicit-CKs and NTKs depend on the activation function and initial weight variances, but *only* via a system of four nonlinear equations. As a direct (and important!) consequence of our theoretical analysis, we demonstrate that (as shallow as) a two-hidden-layer explicit NN with well-designed activations can share the same CK or NTK eigenspectra with a given implicit NN. These findings offer practical benefits and allow for the design of memory-efficient explicit NNs that match implicit NNs' performance without incurring the computational overhead of fixed-point iterations. The proposed theory is supported by empirical results on both synthetic and real-world datasets.

## 1 Introduction

Recently, a novel approach in neural network design has gained prominence in the form of Implicit Neural Networks (NNs) (Bai et al., 2019; El Ghaoui et al., 2021). Implicit NNs introduce a paradigm shift by resembling an infinite-depth weight-shared model with input-injection. In contrast to traditional explicit NNs, such as multi-layer perceptrons (MLPs), recurrent neural networks (RNNs), and residual networks (ResNets), implicit NNs derive features by directly solving for the fixed point. This fixed point represents an equilibrium state in the neural network's computation, bypassing conventional layer-by-layer forward propagation. Additionally, implicit NNs offer a notable advantage, as gradients are analytically computed solely through the fixed point via implicit differentiation. Consequently, the training process for implicit NNs requires only constant memory.

Implicit NNs have demonstrated remarkable performance across a variety of applications, including computer vision (Bai et al., 2020; Xie et al., 2022), natural language processing (Bai et al., 2019), neural rendering (Huang et al., 2021), and solving inverse problems (Gilton et al., 2021). Despite the empirical success achieved by implicit NNs, our theoretical understanding of these models is still limited. As a telling example, it remains unclear whether the training and/or generalization properties of implicit NNs can be connected to those of explicit NNs. Bai et al. (2019) demonstrates that any deep explicit NN can be reformulated as an implicit NN with carefully-designed weight reparameterization. However, many questions such as "whether general implicit NNs have advantages over explicit NNs" or "whether an equivalent explicit NN *always* exists for a given implicit NN", remain largely open. Novel insights into the aforementioned questions are strongly desired since implicit NNs incur significantly higher computational costs than explicit NNs during training and inference, as a consequence of their reliance on iterative solutions to fixed points when computing features or gradients. These iterative solutions involve repeatedly refining computations until they converge to equilibrium states, which is in general computationally intensive. As such, finding explicit shallow NNs that can mimic the behavior of implicit NNs is of great practical significance.

In this paper, building upon recent advances in random matrix theory (RMT), we investigate the high-dimensional eigenspectral behavior of implicit NN models, by focusing on a typical implicit NN, the deep equilibrium model (DEQ) (Bai et al., 2019). We perform a fine-grained asymptotic analysis on the eigenspectra of neural tangent kernels (NTKs) and conjugate kernels (CKs) (Jacot et al., 2018) of implicit NNs, which serve as powerful analytical tools for assessing the convergence and generalization properties of sufficiently wide NNs. For input data following a $K$-class Gaussian mixture model (GMM), we show, in the high-dimensional regime where the data dimension $p$ and their size $n$ are both large and comparable, that the Implicit-CKs and NTKs can be evaluated via more accessible random matrix models that *only* depend on the variance parameter and the activation function via *four* scalar parameters. And possibly more surprisingly, these high-dimensional "proxies" of Implicit-CKs and NTKs have consistent forms with those of explicit NNs established previously in (Ali et al., 2022; Gu et al., 2022).

Inspired by this observation, we establish the high-dimensional "equivalence" between implicit and explicit NNs by matching the key designing parameters of the two nets. In particular, our results reveal that a *two-hidden-layer* explicit NN with carefully designed activations is bound to the same CK or NTK eigenspectra as a given implicit DEQ model, the depth of the latter is essentially *infinite*. Furthermore, in the case of implicit NNs with even or piecewise linear activations like *Tanh or ReLU*, our findings show that a *single-hidden-layer* explicit NN with a thoughtfully designed Leaky ReLU activation exhibits the same CK or NTK eigenspectra. This implies that, at least for GMM data, one can design an equivalent *shallow* explicit NN, which requires the same amount of memory for training and inference as implicit NNs, but avoids the significant computational overhead arising from fixed-point iterations. Despite our theoretical results are derived for GMM data, we observe an unexpected close match between our theory and the empirical results on real-world datasets.

## 1.1 RELATED WORKS

Here, we provide a brief review of related previous efforts.

**Neural tangent kernels.** Neural Tangent Kernel (NTK), initially proposed by Jacot et al. (2018), examines the behavior of wide deep neural networks (DNNs) when trained using gradient descent with small steps. In short, NTK is a specific kernel defined in the context of DNN. During (gradient descent) training, the network parameters change and the NTK also evolves over time. It has been shown by Jacot et al. (2018) and follow-up works that for sufficiently wide DNNs trained on gradient descent with small learning rate, (i) the NTK is approximately constant after initialization and (ii) running gradient descent to update the network parameters is *equivalent* to kernel gradient descent with the NTK. This duality allows one to assess the training dynamics, generalization, and predictions of wide DNNs as *closed-form* expressions involving eigenvalues and eigenvectors of the NTK (Bartlett et al., 2021, Section 6). Originally developed for fully-connected networks, the NTK framework has since then been expanded to convolutional (Arora et al., 2019), graph (Du et al., 2019), and recurrent (Alemohammad et al., 2020) network settings.

**Over-parameterized implicit neural networks.** Feng and Kolter (2020) extended previous NTK studies to implicit NN models and derived the exact expressions of the CK and NTK of ReLU implicit NNs. This study particularly asserts that (i) the NTK of implicit NNs is equivalent to the corresponding weight-untied models in infinitely wide regime and (ii) implicit NNs have non-degenerate NTKs even in the infinite depth. However, the connections between implicit and explicit NTKs remain unexplored. Here we perform a fine-grained analysis on the CK and NTK of implicit NNs and establish the equivalence between implicit DEQ model and explicit NN model in high-dimensional scenarios. Also, while training dynamics (and global convergence) of over-parameterized Implicit NNs have been investigated in previous works (Gao et al., 2022; Gao and Gao, 2022; Ling et al., 2023; Truong, 2023) in the regime of NTKs, it remain unclear what distinguishes the training dynamic of implicit NNs from that of explicit NNs. Moreover, many previous works (Micaelli et al., 2023; Fung et al., 2022; Bai et al., 2022; Ramzi et al., 2021; Bai et al., 2021) have focused on accelerating the training and inference of implicit neural networks. However, these efforts primarily concentrate on developing fast algorithms for implicit differentiation or fixed-point iterations within implicit networks, rather than exploring the connections between implicit and explicit networks.

**Random matrix theory and neural networks.** Random matrix theory (RMT) has emerged as a versatile and potent tool for evaluating the behavior of large-scale systems characterized by a substantial "degree of freedom." Its application has been increasingly embraced in the realm of NN analysis, spanning shallow (Pennington and Worah, 2017; Liao and Couillet, 2018b;a) and deep (Benigni and Péché, 2019; Fan and Wang, 2020; Pastur, 2022; Pastur and Slavin, 2023) models, as well as homogeneous (*e.g.,* standard normal) (Pennington and Worah, 2017; Mei and Montanari, 2022) and mixture-type datasets (Liao and Couillet, 2018b; Ali et al., 2022; Gu et al., 2022). From a technical perspective, the most relevant papers are Ali et al. (2022) and Gu et al. (2022), in which the eigenspectra of CK and NTK were evaluated, for explicit single-hidden-layer NN in Ali et al. (2022) and explicit deep NNs with multi (but finite) layer in Gu et al. (2022). Here, we extend previous analysis to implicit NNs with an effectively *infinite* depth, and establish an equivalence between implicit and explicit NNs.

## 1.2 OUR CONTRIBUTIONS

Our contributions are summarized as follows.

(1) We provide, in Theorems 1 and 2 respectively, for high-dimensional GMM data, precise eigenspectral characterizations of CK and NTK matrices of Implicit NNs; we particularly show that Implicit-CKs and NTKs *only* depend on the variance parameter and the activation function via a few scalar parameters.

(2) We establish, in Corollaries 1 and 2, by matching the key designing parameters derived in Theorems 1 and 2, the equivalence between CKs (and NTKs) of a given implicit DEQ model and *shallow* explicit NN model with carefully-designed activations.

(3) We present empirical evidence using (not-so) wide DNNs trained on synthetic Gaussian datasets and real-world datasets such as MNIST (LeCun et al., 1998), Fashion-MNIST (Xiao et al., 2017), and CIFAR-10 (Krizhevsky, 2009). Our results illustrate that the proposed shallow and carefully-designed explicit NNs achieve comparable performance with respect to implicit NNs, while incurring reduced computational overhead.

## 2 PRELIMINARIES

**Notations.** We use $\mathcal{N}(0, \boldsymbol{I})$ to denote the standard Gaussian distribution. For a vector $\boldsymbol{v}$, $\|\boldsymbol{v}\|$ is the Euclidean norm of $\boldsymbol{v}$. For a matrix $\boldsymbol{A}$, we use $\boldsymbol{A}_{ij}$ denote its $(i, j)$-th entry, and use $\|\boldsymbol{A}\|_F$ to denote the Frobenius norm and $\|\boldsymbol{A}\|$ to denote the operator norm. We use $\odot$ to denote the Hadamard product. We let $\mathcal{O}(\cdot)$ and $\Omega(\cdot)$ denote standard Big-O and Big-Omega notations, respectively. We let $\mathcal{O}_{\|\cdot\|}(n^{-1/2})$ denotes matrices of spectral norm order $\mathcal{O}(n^{-1/2})$.

**Implicit NNs.** In this paper, we focus on the deep equilibrium model (DEQ) Bai et al. (2019). Let $\boldsymbol{X} = [\boldsymbol{x}_1, \cdots, \boldsymbol{x}_n] \in \mathbb{R}^{p \times n}$ denote the input data. We define a vanilla DEQ with the transform at the $l$-th layer as

$$\boldsymbol{h}_i^{(l)} = \sqrt{\frac{\sigma_a^2}{m}} \boldsymbol{A} \boldsymbol{z}_i^{(l-1)} + \sqrt{\frac{\sigma_b^2}{m}} \boldsymbol{B} \boldsymbol{x}_i, \quad \boldsymbol{z}_i^{(l)} = \phi(\boldsymbol{h}_i^{(l)}) \tag{1}$$

where $\boldsymbol{A} \in \mathbb{R}^{m \times m}$ and $\boldsymbol{B} \in \mathbb{R}^{m \times p}$ are weight matrices, $\sigma_a, \sigma_b \in \mathbb{R}$ are constants, $\phi$ is an element-wise activation, $\boldsymbol{h}_i^{(l)}$ is the pre-activation and $\boldsymbol{z}_i^{(l)} \in \mathbb{R}^m$ is the output feature of the $l$-th hidden layer corresponding to the input data $\boldsymbol{x}_i$. The output of the last hidden layer is defined by $\boldsymbol{z}_i^* \triangleq \lim_{l \to \infty} \boldsymbol{z}_i^{(l)}$ and we denote the corresponding pre-activation by $\boldsymbol{h}_i^*$. Note that $\boldsymbol{z}_i^*$ can be calculated by directly solving for the equilibrium point of the following equation

$$\boldsymbol{z}_i^* = \phi\left(\sqrt{\frac{\sigma_a^2}{m}} \boldsymbol{A} \boldsymbol{z}_i^* + \sqrt{\frac{\sigma_b^2}{m}} \boldsymbol{B} \boldsymbol{x}_i\right). \tag{2}$$

Moreover, we define the network's prediction as $f(\boldsymbol{x}_i) = \boldsymbol{a}^\top \boldsymbol{z}_i^*$, for $i \in [n]$, where $\boldsymbol{a} \in \mathbb{R}^m$. We are interested in the associated conjugate kernel and the neural tangent kernel (Implicit-CK and Implicit-NTK, for short) of implicit neural networks defined in Eq. (2). According to the results

in (Feng and Kolter, 2020, Theorem 2), the corresponding Implicit-CK takes the following form

$$\boldsymbol{G}^* = \lim_{l \to \infty} \boldsymbol{G}^{(l)}, \tag{3}$$

where the $(i,j)$-th entry of $\boldsymbol{G}^{(l)}$ is defined recursively as $\boldsymbol{G}_{ij}^{(0)} = \boldsymbol{x}_i^\top \boldsymbol{x}_j$ and[1]

$$\boldsymbol{G}_{ij}^{(l)} = \sigma_a^2 \mathbb{E}_{(\mathrm{u},\mathrm{v}) \sim \mathcal{N}(0, \boldsymbol{\Lambda}_{ij}^{(l)})}[\phi(\mathrm{u})\phi(\mathrm{v})] + \sigma_b^2 \boldsymbol{x}_i^\top \boldsymbol{x}_j, \quad \boldsymbol{\Lambda}_{ij}^{(l)} = \begin{bmatrix} \boldsymbol{G}_{ii}^{(l-1)} & \boldsymbol{G}_{ij}^{(l-1)} \\ \boldsymbol{G}_{ji}^{(l-1)} & \boldsymbol{G}_{jj}^{(l-1)} \end{bmatrix}, \quad l \geq 1. \tag{4}$$

The Implicit-NTK is defined as $\boldsymbol{K}^* = \lim_{l \to \infty} \boldsymbol{K}^{(l)}$ whose the $(i,j)$-th entry is defined as

$$\boldsymbol{K}_{ij}^{(l)} = \sum_{h=1}^{l+1} \left( \boldsymbol{G}_{ij}^{(h-1)} \prod_{h'=h}^{l+1} \dot{\boldsymbol{G}}_{ij}^{(h')} \right), \tag{5}$$

with $\dot{\boldsymbol{G}}_{ij}^{(l)} = \sigma_a^2 \mathbb{E}_{(\mathrm{u},\mathrm{v}) \sim \mathcal{N}(0, \boldsymbol{\Lambda}_{ij}^{(l)})}[\phi'(\mathrm{u})\phi'(\mathrm{v})]$. The limit of Implicit-NTK is

$$\boldsymbol{K}_{ij}^* \equiv \frac{\boldsymbol{G}_{ij}^*}{1 - \dot{\boldsymbol{G}}_{ij}^*}. \tag{6}$$

In this paper, we focus on fully-connected implicit DEQs under the following assumptions regarding the weights and activation functions.

**Assumption 1** (Initialization). *The random matrices $\boldsymbol{A} \in \mathbb{R}^{m \times m}$ and $\boldsymbol{B} \in \mathbb{R}^{m \times p}$ are independent and have* i.i.d. *entries of zero mean, unit variance, and finite fourth-order moment. We consider, without loss of generality, that $\sigma_a^2 + \sigma_b^2 = 1$.*

**Assumption 2** (Activation functions). *The activation function $\phi$ is a $L_1$-Lipschitz function, and at least four-times differentiable with respect to standard normal measure,* i.e., $\max_{k \in \{0,1,2,3,4\}} |\mathbb{E}[\phi^{(k)}(\xi)]| < C_k$ *where $C_k$ is some universal constant and $\xi \sim \mathcal{N}(0,1)$.*

Using the Gaussian integration by parts formula, one has $\mathbb{E}[\phi'(\xi)] = \mathbb{E}[\xi\phi(\xi)]$ for $\xi \sim \mathcal{N}(0,1)$, as long as the right-hand side expectation exists. As a result, Assumption 2 applies for commonly used piecewise linear activations, *e.g.* ReLU.

**The existence and the uniqueness of Implicit-CKs and Implicit-NTKs.** Our formulation requires the existence and the uniqueness of the Implicit-CK $\boldsymbol{G}^*$ and the Implicit-NTK $\boldsymbol{K}^*$. By Eq. (6), we find that the existence and the uniqueness of the Implicit-NTK is determined by those of the corresponding Implicit-CK. Moreover, we note that the $(i,j)$-th entry of is $\boldsymbol{G}^{(l)}$ determined by the inner product $(\boldsymbol{z}_i^{(l-1)})^\top \boldsymbol{z}_j^{(l-1)}$, for $\boldsymbol{z}_i^{(l)}$ defined in Eq. (1), which implies that the existence and the uniqueness of $\boldsymbol{G}^*$ can be guaranteed by those of $\boldsymbol{z}_i^*$, for $i \in [n]$. Therefore, we propose to ensure the existence and uniqueness of $\boldsymbol{z}^*$ by imposing the following condition, there by ensuring those of the Implicit-CK and the Implicit-NTK.

**Condition 1.** *The variance parameter defined in Eq. (2) satisfies $\sigma_a^2 < \frac{1}{4L_1^2}$.*

For Condition 1, we apply the consequence of standard bounds concerning the singular values of random matrices (Vershynin, 2018), namely, it holds that $\|\boldsymbol{A}\| \leq 2$ with exponentially high probability. Furthermore, by noting that $\phi(\cdot)$ is $L_1$-Lipschitz, one can easily demonstrate that the transformation in Eq. (1) is a *contractive* mapping, and thus ensuring the existence of the unique fixed point of $\boldsymbol{z}^*$.

**Gaussian mixture data.** We consider $n$ data vectors $\boldsymbol{x}_1, \cdots, \boldsymbol{x}_n \in \mathbb{R}^p$ independently drawn from one of the $K$-class Gaussian mixture $\mathcal{C}_1, \cdots, \mathcal{C}_K$ and denote $\boldsymbol{X} = [\boldsymbol{x}_1, \cdots, \boldsymbol{x}_n] \in \mathbb{R}^{p \times n}$, with class $\mathcal{C}_a$ having cardinality $n_a$, *i.e.*, for $\boldsymbol{x}_i \in \mathcal{C}_a$, we have

$$\boldsymbol{x}_i \sim \mathcal{N}(\boldsymbol{\mu}_a/\sqrt{p}, \boldsymbol{C}_a/p), \tag{7}$$

---

[1] Note that the expectation is conditioned on the input data, and is taken with respect to the random weights.

**Assumption 3** (High-dimensional asymptotics). *We assume that, as $n \to \infty$, for $a \in \{1, \cdots, K\}$, (i) $p/n \to c \in (0, \infty)$ and $n_a/n \to c_a \in (0,1)$; (ii) $\|\boldsymbol{\mu}_a\| = \mathcal{O}(1)$; (iii) for $\boldsymbol{C}^\circ \equiv \sum_{a=1}^K \frac{n_a}{n} \boldsymbol{C}_a$ and $\boldsymbol{C}_a^\circ \equiv \boldsymbol{C}_a - \boldsymbol{C}^\circ$, we have $\|\boldsymbol{C}_a\| = \mathcal{O}(1)$, $\mathrm{tr}\, \boldsymbol{C}_a^\circ = \mathcal{O}(\sqrt{p})$ and $\mathrm{tr}(\boldsymbol{C}_a \boldsymbol{C}_b) = \mathcal{O}(p)$ for $a, b \in \{1, \cdots, K\}$; and (iv) $\tau_0 \equiv \sqrt{\mathrm{tr}\, \boldsymbol{C}^\circ / p}$ converges in $(0, \infty)$.*

**Remark 1** (On GMM data and Assumption 3). Note that the Gaussian mixture model defined in Eq. (7) is nothing but standard multivariate Gaussian distribution $\mathcal{N}(\boldsymbol{\mu}_a, \boldsymbol{C}_a)$ normalized by $1/\sqrt{p}$. This normalization is commonly used in the literature of high-dimensional statistics and over-parameterized DNNs and ensures, together with Assumption 3, that the data vectors have bounded norms $\|\boldsymbol{x}_i\| = \mathcal{O}(1)$ in the $n, p \to \infty$ limit. The high-dimensional asymptotics as $n, p \to \infty$ with $p/n \to c \in (0, \infty)$ in Assumption 3 does *not* demand that $n, p$ be growing but merely that they be *both* large. And the obtained approximations error in Theorems 1 and 2 would be of order $\mathcal{O}(n^{-1/2})$ or $\mathcal{O}(p^{-1/2})$. The GMM in Eq. (7) and the high-dimensional (non-trivial) classification setting in Assumption 3 have been extensively studied in the literature for various ML methods ranging from kNN, LDA, spectral clustering, SVM, to shallow neural networks, see for example (Louart et al., 2018; Couillet and Liao, 2022; Couillet et al., 2018; Dobriban and Wager, 2018) as well as (Blum et al., 2020, Section 2), and have led to, e.g., a thorough theoretical understanding of the so-called "double descent" curves for over-parameterized ML models (Mei and Montanari, 2022).

## 3 MAIN RESULTS

We present in Section 3.1 our main technical results on the high-dimensional characterization of CK and NTK matrices of implicit NNs, in Theorems 1 and 2, respectively. We show in Section 3.2 that the proposed theoretical analysis allows to construct, for a given implicit DEQ model, an equivalent and not-so-deep *explicit* NN model (having at most two hidden layers) that shares the same CK or NTK eigenspectral behavior as the implicit NN.

### 3.1 HIGH-DIMENSIONAL CHARACTERIZATION OF IMPLICIT-CK AND NTK MATRICES

For ease of presentation, let us first define some useful quantities. For Gaussian mixture data defined in Eq. (7), we denote

$$\boldsymbol{J} \equiv [\boldsymbol{j}_1, \cdots, \boldsymbol{j}_K] \in \mathbb{R}^{n \times K}, \quad \boldsymbol{j}_a \in \mathbb{R}^n, \tag{8}$$

with label vector $[\boldsymbol{j}_a]_i = \delta_{\boldsymbol{x}_i \in \mathcal{C}_a}$ of class $\mathcal{C}_a$, $a \in \{1, \ldots, K\}$, and rows of $\boldsymbol{J}$ the standard one-hot-encoded label vectors in $\mathbb{R}^K$. We define the second-order data fluctuation vector as

$$\boldsymbol{\psi} \equiv \{\|\boldsymbol{x}_i - \mathbb{E}[\boldsymbol{x}_i]\|^2 - \mathbb{E}[\|\boldsymbol{x}_i - \mathbb{E}[\boldsymbol{x}_i]\|^2]\}_{i=1}^n \in \mathbb{R}^n, \tag{9}$$

and use

$$\boldsymbol{T} = \{\mathrm{tr}\, \boldsymbol{C}_a \boldsymbol{C}_b / p\}_{a,b=1}^K \in \mathbb{R}^{K \times K}, \quad \boldsymbol{t} = \{\mathrm{tr}\, \boldsymbol{C}_a^\circ / \sqrt{p}\} \in \mathbb{R}^K, \tag{10}$$

to denote the second-order discriminative statistics of the Gaussian mixture in Eq. (7). These quantities, as we shall see below, will be consistently used in the high-dimensional characterizations of CK and NTK matrices, for both implicit and explicit NN models.

**Condition 2.** *The activation function $\phi$ satisfies $\mathbb{E}[(\phi^2(\tau \xi))''] < L_2$ for $\xi \sim \mathcal{N}(0,1)$ and some universal constant $L_2 > 0$, and the variance parameter defined in Eq. (2) satisfies $\sigma_a^2 < 2/L_2$.*

Define $\tau_*$ the fixed point of the following equation

$$\tau_* = \sqrt{\sigma_a^2 \mathbb{E}\left[\phi^2(\tau_* \xi)\right] + (1 - \sigma_a^2)\tau_0^2}, \quad \xi \sim \mathcal{N}(0,1), \tag{11}$$

the existence and uniqueness of which is ensured under Assumptions 1-2, per the following remark.

**Remark 2** (Existence and uniqueness of $\tau_*$). It can be checked that for any given $\tau_0 > 0$ and variance parameter $\sigma_a$ that satisfies Condition 2, the right-hand side of Eq. (11) constitutes a *contractive mapping*, thereby ensuring the existence of a unique fixed point $\tau_*$. Please see Lemma. A.1 in the supplementary material for a detailed proof of this fact.

With these notations at hand, we are ready to present our first result on the high-dimensional characterization of the CK matrices for implicit NNs, the proof of which is given in Appendix B of the supplementary material.

**Theorem 1** (Asymptotic approximation of Implicit-CKs). *Let Assumptions 1-3 hold, and let the activation $\phi(\cdot)$ be "centred" such that $\mathbb{E}[\phi(\tau_*\xi)] = 0$ for $\xi \sim \mathcal{N}(0,1)$ and $\tau_*$ defined in Eq. (11). Further assume that the variance parameter $\sigma_a$ satisfies Conditions 1 and 2. Then, the Implicit-CK matrix $\mathbf{G}^*$ defined in Eq. (3) can be well approximated, in an operator norm sense, by the random matrix $\overline{\mathbf{G}}$ as*

$$\left\| \mathbf{G}^* - \overline{\mathbf{G}} \right\| = \mathcal{O}\left(n^{-1/2}\right), \quad \overline{\mathbf{G}} \equiv \alpha_{*,1}\mathbf{X}^\top\mathbf{X} + \mathbf{V}\mathbf{C}_*\mathbf{V}^\top + (\tau_*^2 - \tau_0^2\alpha_{*,1} - \tau_0^4\alpha_{*,3})\mathbf{I}_n, \quad (12)$$

*where*

$$\mathbf{V} = [\mathbf{J}/\sqrt{p},\ \boldsymbol{\psi}] \in \mathbb{R}^{n\times(K+1)}, \quad \mathbf{C}_* = \left[\begin{array}{cc} \alpha_{*,2}\mathbf{t}\mathbf{t}^\top + \alpha_{*,3}\mathbf{T} & \alpha_{*,2}\mathbf{t} \\ \alpha_{*,2}\mathbf{t}^\top & \alpha_{*,2} \end{array}\right] \in \mathbb{R}^{(K+1)\times(K+1)}, \quad (13)$$

*with non-negative scalars $\alpha_{*,1}, \alpha_{*,2}, \alpha_{*,3}, \alpha_{*,4} \geq 0$ defined, for $\xi \sim \mathcal{N}(0,1)$, as*

$$\alpha_{*,1} = \frac{1 - \sigma_a^2}{1 - \sigma_a^2\mathbb{E}[\phi'(\tau_*\xi)]^2}, \qquad\qquad \alpha_{*,2} = \frac{\sigma_a^2\mathbb{E}[\phi''(\tau_*\xi)]^2}{4\left(1 - \sigma_a^2\mathbb{E}[\phi'(\tau_*\xi)]^2\right)}\alpha_{*,4}^2, \qquad (14)$$

$$\alpha_{*,3} = \frac{\sigma_a^2\mathbb{E}[\phi''(\tau_*\xi)]^2}{2\left(1 - \sigma_a^2\mathbb{E}[\phi'(\tau_*\xi)]^2\right)}\alpha_{*,1}^2, \qquad \alpha_{*,4} = \frac{1 - \sigma_a^2}{1 - \frac{\sigma_a^2}{2}\mathbb{E}[(\phi^2(\tau_*\xi))'']}. \qquad (15)$$

Theorem 1 reveals the surprising fact that, for high-dimensional GMM data in Eq. (7), the implicit CK matrix $\mathbf{G}^*$, despite its mathematically involved form (as the fixed point of the recursion) in Eq. (3), is approximately equivalent to a much more simple matrix. This "equivalent" CK matrix $\overline{\mathbf{G}}$,

    (i) depends, as expected, on the input GMM data ($\mathbf{X}$), their class structure ($\mathbf{J}$) and statistics ($\mathbf{t}$ and $\mathbf{T}$), but in a rather explicit fashion; and

    (ii) is *independent* of the distribution of the weight matrices $\mathbf{A}$ and $\mathbf{B}$; and

    (iii) depends on $\sigma_a^2$ and the activation $\phi$ *only* via four scalars $\alpha_{*,1}, \alpha_{*,2}, \alpha_{*,3}$ and $\tau_*$.

A similar result can be derived for the NTK matrices of implicit NNs and is given as follows.

**Theorem 2** (Asymptotic approximation of Implicit-NTKs). *Let Assumptions 1-3 hold, and let the activation $\phi(\cdot)$ be "centred" such that $\mathbb{E}[\phi(\tau_*\xi)] = 0$ for $\xi \sim \mathcal{N}(0,1)$ and $\tau_*$ defined in Eq. (11). Further assume that the variance parameter $\sigma_a$ satisfies Conditions 1 and 2. Then, the Implicit-NTK matrix $\mathbf{K}^*$ defined in Eq. (6) can be well approximated, in an operator norm sense, by the random matrix $\overline{\mathbf{K}}$ as*

$$\left\| \mathbf{K}^* - \overline{\mathbf{K}} \right\| = \mathcal{O}\left(n^{-1/2}\right), \quad \overline{\mathbf{K}} \equiv \beta_{*,1}\mathbf{X}^\top\mathbf{X} + \mathbf{V}\mathbf{D}_*\mathbf{V}^\top + (\kappa_*^2 - \tau_0^2\beta_{*,1} - \tau_0^4\beta_{*,3})\mathbf{I}_n, \quad (16)$$

*where $\kappa_*^2 = \tau_*^2/\left(1 - \sigma_a^2\mathbb{E}\left[\phi'(\tau_*\xi)^2\right]\right)$ and*

$$\mathbf{V} = [\mathbf{J}/\sqrt{p},\ \boldsymbol{\psi}] \in \mathbb{R}^{n\times(K+1)}, \quad \mathbf{D}_* = \left[\begin{array}{cc} \beta_{*,2}\mathbf{t}\mathbf{t}^\top + \beta_{*,3}\mathbf{T} & \beta_{*,2}\mathbf{t} \\ \beta_{*,2}\mathbf{t}^\top & \beta_{*,2} \end{array}\right] \in \mathbb{R}^{(K+1)\times(K+1)}, \quad (17)$$

*with non-negative scalars $\beta_{*,1}, \beta_{*,2}, \beta_{*,3} \geq 0$ defined, for $\xi \sim \mathcal{N}(0,1)$, as*

$$\beta_{*,1} = \frac{\alpha_{*,1}}{1 - \sigma_a^2\mathbb{E}[\phi'(\tau_*\xi)]^2}, \beta_{*,2} = \frac{\alpha_{*,2}}{1 - \sigma_a^2\mathbb{E}[\phi'(\tau_*\xi)]^2}, \beta_{*,3} = \frac{\alpha_{*,3} + \beta_{*,1}\sigma_a^2\mathbb{E}[\phi''(\tau_*\xi)]^2\alpha_{*,1}}{1 - \sigma_a^2\mathbb{E}[\phi'(\tau_*\xi)]^2}.$$

We refer the readers to Appendix C of the supplementary material for the proof of Theorem 2. Theorem 2 tells us that the NTK matrices of implicit NNs take a similar form as the CK matrices, and (approximately) depend on $\sigma_a$ and the activation via $\beta_{*,1}, \beta_{*,2}, \beta_{*,3}$ and $\kappa_*$.

**Remark 3** (On centered activation). *Given any activation function $\tilde{\phi}(\cdot)$ that satisfies Assumption 2, a centered activation $\phi$ can be obtained by simplifying subtracting a constant as $\phi(x) = \tilde{\phi}(x) - \mathbb{E}[\tilde{\phi}(\tau_*x)]$ with $\tau_* = \sqrt{\sigma_a^2\mathbb{E}[(\tilde{\phi}(\tau_*\xi) - \mathbb{E}[\tilde{\phi}(\tau_*\xi)])^2] + (1 - \sigma_a^2)\tau_0^2}$.*

## 3.2 THE EQUIVALENCE BETWEEN IMPLICIT AND EXPLICIT NNS IN HIGH DIMENSIONS

Implicit NNs are known, per its definition in Eq. (2), to be formally equivalent to *infinitely* deep *explicit* NN model (Bai et al., 2020; Xie et al., 2022). In the sequel, we show how our proposed theoretical results in Theorems 1 and 2 allow one to construct *explicit* and *not-so-deep* NN models that are "equivalent" to a given *implicit* DEQ model, in the sense that the CK and/or NTK matrices of the two networks are close in operator norm for $n, p$ large.

Consider the following *finitely* deep *explicit* NN model having $L$ layers,

$$\boldsymbol{x}_i^{(l)} = \frac{1}{\sqrt{m_l}} \sigma_l(\boldsymbol{W}_l \boldsymbol{x}_i^{(l-1)}), \quad \text{for } l = 1, \cdots, L, \tag{18}$$

where $\boldsymbol{W}_l \in \mathbb{R}^{m_l \times m_{l-1}}$ are weight matrices and $\sigma_l$ are element-wised activation functions. We denote $\boldsymbol{X}^{(l)} = \frac{1}{\sqrt{m_l}} \sigma_l(\boldsymbol{W}_l \boldsymbol{X}^{(l-1)})$ the representations of the input data matrix $\boldsymbol{X} \in \mathbb{R}^{p \times n}$ at layer $l \in \{1, \cdots, L\}$. For the fully-connected explicit NN model given in Eq. (18), the corresponding Explicit-CK matrix $\boldsymbol{\Sigma}^{(l)}$ at layer $l$ is defined as (Fan and Wang, 2020)

$$\boldsymbol{\Sigma}_{ij}^{(l)} = \mathbb{E}_{u,v}[\sigma_l(u)\sigma_l(v)], \quad \text{with } (u, v) \sim \mathcal{N}\left(0, \begin{bmatrix} \boldsymbol{\Sigma}_{ii}^{(l-1)} & \boldsymbol{\Sigma}_{ij}^{(l-1)} \\ \boldsymbol{\Sigma}_{ji}^{(l-1)} & \boldsymbol{\Sigma}_{jj}^{(l-1)} \end{bmatrix}\right), \tag{19}$$

and the Explicit-NTK matrix $\boldsymbol{\Theta}^{(l)}$ at layer $l$ is defined as

$$\boldsymbol{\Theta}^{(l)} = \boldsymbol{\Sigma}^{(l)} + \boldsymbol{\Theta}^{(l-1)} \odot \dot{\boldsymbol{\Sigma}}^{(l)}, \quad \boldsymbol{\Theta}^{(0)} = \boldsymbol{\Sigma}^{(0)} = \boldsymbol{X}^\top \boldsymbol{X}, \tag{20}$$

where $\dot{\boldsymbol{\Sigma}}^{(l)}$ denotes the CK matrix with activation $\sigma_l'$ instead of $\sigma_l$ with $\dot{\boldsymbol{\Sigma}}_{ij}^{(l)} = \mathbb{E}_{u,v}[\sigma_l'(u)\sigma_l'(v)]$. As in Assumption 1, we assume that weight matrices $\boldsymbol{W}_l$s have *i.i.d.* entries of zero mean, unit variance, and finite fourth-order moment.

The high-dimensional behaviors of both the CK and NTK matrices for the fully-connected explicit NN model in Eq. (18) have been recently studied in (Gu et al., 2022) using RMT techniques.

In this vein, our Theorems 1 and 2 apply to make an *explicit* connection between implicit and explicit NN models. In the following result, we show how to construct a *two-hidden-layer* explicit NN with polynomial activation that admits approximately the same CK as a given implicit DEQ.

**Corollary 1** (Equivalent poly-ENN). *For a given fully-connected implicit NN (denoted* INN*) with centered activation such that* $\mathbb{E}[\phi(\tau_*\xi)] = 0$ *for* $\xi \sim \mathcal{N}(0,1)$ *and* $\tau_*$ *in Eq. (11), one is able to construct a two-hidden-layer "equivalent" explicit NN having quadratic polynomial activations:* $\sigma_l(x) = a_l x^2 + b_l x + c_l, l = 1, 2$ *(denoted* poly-ENN*), in such a way that the two nets have asymptotically the same CK eigenspectra with* $\|\boldsymbol{G}^* - \boldsymbol{\Sigma}^{(2)}\| = \mathcal{O}(n^{-1/2})$, *by solving a system of polynomial equations (see Appendix D for a detailed exposition).*

*Proof sketch of Corollary 1.* The proof of Corollary 1 starts from the observation that the asymptotic equivalent of the Implicit-CK in Eq. (12) takes a similar form to that of the Explicit-CK as given in (Gu et al., 2022, Theorem 1), with their coefficients determined by activations. Thus, it suffices to choose the activations of poly-ENN such that the corresponding Explicit-CK yields the same coefficients as the Implicit-CK. Please see the complete proof of Corollary 1 in Appendix D. □

Corollary 1 provides explicit connections between INNs and ploy-ENNs, as well as a general recipe to construct a ploy-ENN equivalent to any given INN when measured by their corresponding CK matrices. Results for NTK matrices can be similarly obtained by combining our Theorem 2 and (Gu et al., 2022, Theorem 2) and is thus omitted here.

There is of course nothing special about the choice of polynomial activation in the design of "equivalent" explicit NN models in Corollary 1. In the following result, we show that the large family of implicit NNs (INN) with even or piecewise linear activations can be "imitated" by single-hidden-layer explicit NNs (denoted L-ReLU-ENN) having Leaky ReLU-type activation (see, e.g., the left display of Figure 2 for a visualization). We refer the readers to Appendix E for the proof.

**Corollary 2** (Equivalent L-ReLU-ENN). *For a given fully-connected implicit NN with even or piecewise linear activation that satisfies* $\mathbb{E}[\phi(\tau_*\xi)] = 0$ *for* $\xi \sim \mathcal{N}(0,1)$, *one has that* $\mathbb{E}[\phi''(x)] = 0$

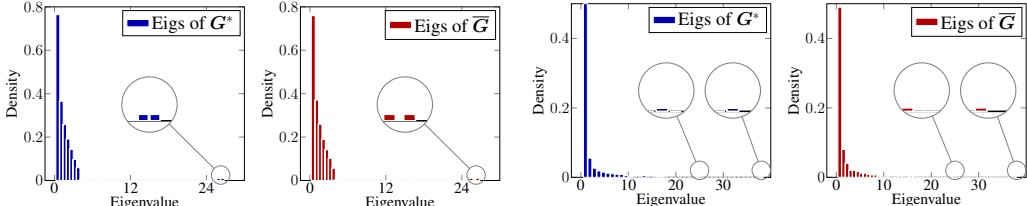

Figure 1: Eigenvalue density of Implicit-CK matrices $G^*$ (**blue**) defined in Eq. (3) (with expectation estimated from 400 independent realizations of random $A$ and $B$) and the asymptotic equivalent matrices $\overline{G}$ (**red**) obtained by Theorem 1. (**Left**): an implicit NN defined in Eq. (2) with Arc-Tanh activation and $\sigma_a^2 = 0.1$, on two-class GMM data, with $p = 1\,000$, $n = 800$, $\boldsymbol{\mu}_a = [\mathbf{0}_{8(a-1)}; 8; \mathbf{0}_{p-8a+7}]$, $\boldsymbol{C}_a = (1 + 8(a-1)/\sqrt{p})\boldsymbol{I}_p$, for $a \in \{1, 2\}$, here $\|G^* - \overline{G}\| \approx 0.12$; and (**Right**): an implicit NN defined in Eq. (2) with Tanh activation and $\sigma_a^2 = 0.1$, on two-class MNIST data (number 6 versus number 8), with $p = 784$, $n = 6\,000$, for which $\|G^* - \overline{G}\| \approx 1.57$.

and one is able to construct a single-hidden-layer *"equivalent"* explicit NN model having biased *Leaky ReLU activation:*

$$\varphi(x) \equiv \max(ax, bx) - \frac{a - b}{\sqrt{2\pi}}\tau_0, \tag{21}$$

*in such a way that the two nets have asymptotically the same CK eigenspectra with* $\|G_\varphi^* - \boldsymbol{\Sigma}^{(1)}\| = \mathcal{O}(n^{-1/2})$, *where* $a > b > 0$ *is determined by solving the following equations:*

$$(a - b)^2 = 4\alpha_{*,1}^2, \quad \frac{(\pi - 1)(a^2 + b^2) + ab}{2\pi}\tau_0^2 - \frac{(a + b)^2}{4}\tau_0^2 = \tau_*^2 - \alpha_{*,1}\tau_0^2.$$

## 4 EXPERIMENTS

In this section, we provide numerical experiments on not-so-high-dimensional data to validate our proposed theoretical results. We consider both synthetic Gaussian mixture data and samples drawn from commonly used real-world datasets such as MNIST (LeCun et al., 1998), Fashion-MNIST (Xiao et al., 2017), and CIFAR-10 (Krizhevsky, 2009).

Figure 1 compares the eigenvalues of Implicit-CKs and their high-dimensional approximation from Theorem 1, for both synthetic Gaussian mixture and MNIST data. We observe that the proposed theoretical results, despite derived here for GMM data *and* in the limit of $n, p \to \infty$, provide extremely accurate prediction of the Implicit-CK eigenspectral behavior (i) for not-so-large $n, p$ *and* (ii) possibly surprisingly, also on realistic MNIST data. We conjecture that this is due to a high-dimensional *universal* phenomenon and that our results (on both CK and NTK matrices) hold more generally beyond the GMM setting, say, for data drawn from the family of concentrated random vectors (Ledoux, 2005; Louart and Couillet, 2018). We refer the interested readers to (Couillet and Liao, 2022, Chapter 8) for more discussions on this point.

In Figure 2 we testify the results in Corollary 2 by constructing explicit single-hidden-layer NN models (L-ReLU-ENN) with Leaky ReLU-type activation that are "equivalent" (in the sense of CK) to implicit NN (INN) with ReLU activation. We see that, while the two types of NN models are different in that (i) INN is implicitly defined while L-ReLU-ENN is explicitly defined; and (ii) INN uses ReLU activation while L-ReLU-ENN uses Leaky ReLU activation. Their CK matrices establish a surprisingly close eigenspectral behavior as long as the activation of L-ReLU-ENN is carefully chosen according to our Corollary 2. This observation is again consistent on synthetic GMM *and* realistic MNIST data.

To see whether this high-dimensional "equivalence" between implicit and not-so-deep explicit NN models can be observed more generally across different realistic datasets, we further compare the classification accuracy of implicit and carefully (or-not) designed explicit NNs in Figure 3. Implicit and explicit NNs share the *same* network width $m \in \{32, 64, 128, 256, 512, 1\,024, 2\,048, 4\,096, 8\,192\}$. As $m$ increases, the performance of L-ReLU-ENNs closely matches that of INN, while a noticeable performance gap exists between ReLU-ENN

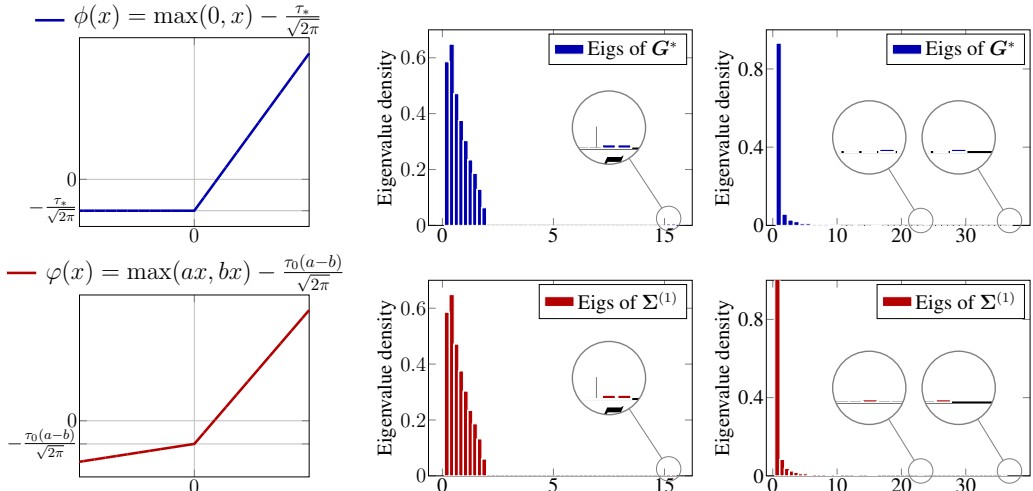

Figure 2: Visualization and eigenvalues of (**top**) Implicit-CKs $G^*$ with (biased) ReLU activation (**blue**) and (**bottom**) Explicit-CKs $\Sigma^{(1)}$ with (biased) Leaky ReLU activation (**red**) defined in Eq. (21). (**Middle**): on two-class GMM data as in Figure 1, with here $\|G^* - \Sigma^{(1)}\| \approx 0.23$; and (**Right**): on two-class MNIST data (number 6 versus number 8), with $p = 784$, $n = 6\,000$, with $\|G^* - \Sigma^{(1)}\| \approx 1.83$. The expectations are estimated from 400 independent realizations.

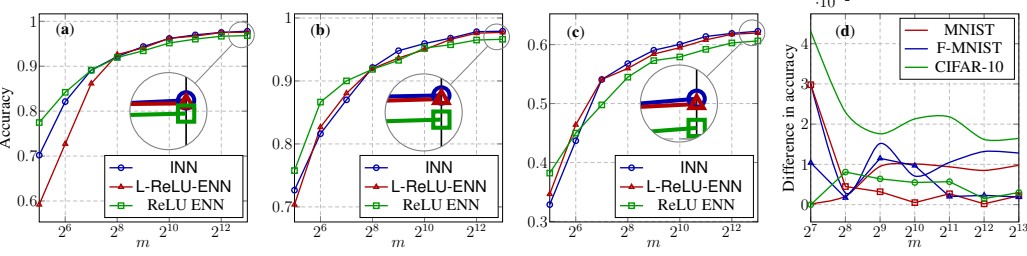

Figure 3: The evolution of classification results *w.r.t* the width $m$ of implicit ReLU NNs (**blue**, INN), the corresponding equivalent single-layer Leaky-ReLU explicit NNs (**red**, L-ReLU-ENN), and ReLU explicit NNs (**green**, ReLU ENN for short) on (**a**) MNIST, (**b**) fashion MNIST, and (**c**) CIFAR-10. For MNIST datasets, raw data are taken as the network input; for CIFAR-10 dataset, flattened output of the 16th convolutional layer of VGG-19 are taken as the network input. The last figure (**d**) visualizes the gap between the performance of INNs and L-ReLU-ENNs, and the gap between the performance of INNs and ReLU ENNs.

and INN. This observation substantiates our theory and underscores the practical advantages of our approach by, e.g., enabling the design of memory-efficient explicit NNs that achieve the performance of implicit NNs without the computational overhead associated with fixed-point iterations.

## 5 CONCLUSION

In this paper, we investigate the connection between implicit NNs and explicit NNs. We employ RMT to analyze the eigenspectra of the NTKs and CKs of implicit NNs. For high-dimensional Gaussian mixture data, we establish asymptotic equivalents for the NTK and CK of implicit NNs. Notably, we reveal that the eigenspectra of the NTK and CK of implicit NNs are determined solely by the variance parameter and the activation function. Based on this observation, we establish the equivalence between implicit NNs and explicit NNs in high dimensions. We propose a method for designing activation functions for explicit neural networks to "match" the spectral behavior of the CK (or NTK) of implicit NNs. Results on synthetic data and real-world data demonstrate that shallow explicit NNs with our theoretically designed activation functions achieve comparable accuracy to implicit NNs, while significantly reducing computational overhead.

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
