# OpenReview forum: "Implicit NNs are Almost Equivalent to Not-so-deep Explicit NNs for High-dimensional Gaussian Mixtures"
_ICLR.cc/2024/Conference — ICLR 2024 Conference Withdrawn Submission_

### Official Review · Reviewer_yQj9 · 2023-10-27

**Soundness:** 3 good
**Presentation:** 3 good
**Contribution:** 3 good
**Rating:** 8
**Confidence:** 2

**Summary:**

The authors investigate a connection between explicit and implicit neural networks. They provide an asymptotic behavior of the implicit CK and NTK matrices. They show that given an implicit NN, there exists a shallow explicit NN with quadratic polynomial activatin functions that is equivalent to the implicit NN.

**Strengths:**

The topic is relevent to the comunity. The paper is well-written.

**Weaknesses:**

Providing more explanations about related work can be helpful for readers. For example, briefly summarizing the result of Gu et al. may be helpful.

**Questions:**

Queations:
How does the assymptotic behavior stated in Theorems 1 and 2 depend on K and p?

Minor comments:
- Eq. (1): There are two "+" symbols.
- In the beginning of Section 3, "Theorem" should be "Theorems".

---

> ### Author Response · Authors · 2023-11-16
>
> We thank the reviewer for his/her feedback and positive assessment of our work.
>
> 1. **more explanation about related work**:
> We thank the reviewer for the valuable suggestion.
> We have added more discussions on the CK and NTK in the related work section, and in (the current) Remark 1 to better motivate the GMM in Eq. (7) and Assumption 3
> A summary of the results in (Gu et al., 2022) is also included.
> Due to space limitation, the summary is placed in **Section D** of the revised supplementary material.
> Moreover, we have fixed the typos pointed out in your minor comments.
>
> 2. **dependence on K and p**. In this paper, we consider $K$ to be fixed and of order $O(1)$ for $n,p$ both large.
> So in particular, the data dimension $p$ is not significantly small than the sample size $n$.
> The assumption of $n,p \to \infty$ with $p/n \to c \in (0,\infty)$ in Assumption 3 is imposed so that the present analysis is performed in a (mathematically rigorously) high-dimensional asymptotic setting.
> The conducted analysis here does **not** demand that $n,p$ be growing variables but merely that they be *both* large (for the large $n,p$ asymptotics to be accurate, say).
> As such, one may freely assume $p$ large but fixed and $n$ growing, without altering the conclusions of our present study (the approximation error for our results would then be of the order $O(p^{-1/2})$ instead of $O(n^{-1/2})$ in Theorem 1 and 2).
> This is in complete adequacy with modern machine learning practice where the data dimensions are generally in hundreds or even thousands.
> The GMM under Assumption 3 has been extensively studied in the literature of high-dimensional ML for a wide class of models ranging from kNN, LDA, SVM, and shallow neural network models, see [D1], Section 2 of [D2], and Section 4.1 of [D3].
> We have added the current **Remark 1** in the revised version for further discussion on this point.
>
>
> [D1] Couillet, R., Liao, Z., & Mai, X. (2018, September). Classification asymptotics in the random matrix regime. In 2018 26th European Signal Processing Conference (EUSIPCO) (pp. 1875-1879). IEEE.
>
> [D2] Blum, A., Hopcroft, J., & Kannan, R. (2020). Foundations of data science. Cambridge University Press.
>
> [D3] Couillet, R., & Liao, Z. (2022). Random matrix methods for machine learning. Cambridge University Press.

---

### Official Review · Reviewer_9wKK · 2023-10-29

**Soundness:** 2 fair
**Presentation:** 3 good
**Contribution:** 2 fair
**Rating:** 3
**Confidence:** 2

**Summary:**

This paper proposes to study implicit Neural Networks (INNs) through the lens of Conjugate Kernels (CKs) and Neural Tangent Kernels (NTKs).
With technical assumptions, among which the fact that the data is sampled from a Gaussian Mixture Model, it is shown that the CK and NTK matrices of INNs are asymptotically determined by a handful of parameters that can be derived from the activation function, the variance of the model weights, the input data and its statistics.
It is also shown that a 2-layer deep explicit neural network can be constructively designed to have CK and NTK matrices that match with that of a given INN.
Finally, experiments on real and synthetic data show the eigenspectral behaviours of the estimated and approximate CK and NTK matrices, as well as the performance of the designed explicit neural networks compared to the original INNs.

**Strengths:**

- the problem tackled is important: trying to understand INNs in a theoretical way is critical to using them at their fullest capability
- experiments on real and synthetic data are important
- assumptions are clearly listed before each theoretical results

**Weaknesses:**

- **Code**: code is not provided to reproduce the experiments. To me it's a huge problem because it is a hinderance to reproducibility and I don't think there are reasons that would prevent the code to be shared.
- **Interest of CK and NTK**: I am mainly familiar with INNs, not so much with CK and NTK. This is why it's not apparent to me simply reading the paper why characterizing CK and NTK is a good proxy to understand the overall behaviour of INNs. I think the sentence trying to explain this is "which serve as powerful analytical tools for assessing the convergence and generalization properties of sufﬁcienlt wide NNs" but it reamins unclear to me. To be perfectly clear, I do think that they could be important, I just don't understand why in this context. The related work part about NTK is also unclear to me.
- **Bias in INNs**: in eq (1), there is no bias as opposed to Feng and Kolter 2020. I think it would be important to comment on why this parameter is gone from the parametrization and how having it would affect the proof.
- **Assumption 3.**: assumption 3 is not commented at all. I think it's super important to comment all of these aspects especially those related to the asymptotic part. For example, the dimensionality of the data increases with the number of data points: this is very unusual and typically breaks the PAC-learning framework. Similarly, asymptotically, the mean of the data is 0 (if I understand correctly), since there is no bias in the formulation of the INNs, this does not cover all cases by a simple translation, so I think it's important to comment on that as well.
- **Explicit NNs**: one of the conclusions I draw from section 3.2 is that it is possible to design explicit NNs that are virtually equivalent to INNs. It is also said that these would be much more efficient computationally than the corresponding INNs: if the widths were roughly equivalent this would be true, but I don't see any mention of this. If they are not, I think it needs to be proved theoretically. In any case, I think some experiments could show this easily by just computing the time of the forward pass.
- **Relevance of the experiments**: several aspects are to me questionnable:
   1. It seems to me that the experiments do not illustrate the result of Theorem 1 or 2 which are asymptotic. Here a single snapshot for a given pair $(p, n)$ is given rather than a plot showing $\|G^\star - \bar{G}\|$ as a function of $n$ even for simulated data. Of course this would be harder for real data since $p$ has to grow with $n$, but it also shows that this assumption is questionnable. Moreover, this would give some context for the values of $\|G^\star - \bar{G}\|$ rather than have absolute values.
   2. I don't understand why the eigenvalues become an important part of the experimental section even though they were not mentioned in the theoretical part. Why can't we just focus on $\|G^\star - \bar{G}\|$? Moreover, the remarks on the comparison of eigenspectral behavior (page 8, second paragraph) are not quantitatively grounded. For example, I could argue that the eigenvalues distributions shown in the right panels of Figure 1 are not similar looking : the second bar in the red plot is much higher than the second bar in the left plot.
   3. It is mentioned that the gap between ReLU-ENN and L-ReLU-ENN is noticeable but it needs a zoom to be seen and no error bars are given to assess this more quantitatively. Moreover, the performance on CIFAR-10 is extremely low on the order of 60%, very far from even basic networks on this problem: this is a very worrying sign that maybe some procedures are not conducted properly (parametrization, optimization, evaluation, ...). At the very least this should be commented, but it shows that the experiments are very far from a practical setting and it bears the question of whether these results would hold on SoTA INNs. I want to clarify that I understand it's not the matter of this paper to beat SoTA in any shape or form, but the numbers on CIFAR-10 are so low that it's an orange flag to me.

Minor:
- there are a lot of typos -> grammarly or Ltex workshop can help reduce these (or even the use of LLMs)
- Since the computational efficiency of INNs is a matter here, I think it would be interesting to cite relevant literature that tries to accelerate the training and inference of such networks (Neural DEQ solver [E], SHINE [D], Jacobian-free backprop [B], warm start [A, C])
- It is said "Following (Feng and Kolter, 2020), we denote the corresponding Implicit-CK as": here I think it would be important to explain that this is not a definition per-se but rather a property of the CK of INNs derived by Feng and Kolter 2020, and also give the initial definition.
- There are a lot of notations, which is very confusing for any reader IMO, I think a lot could be improved by moving the NTK results in the appendix and only mention them in the core text, and move a lot more of the intermediate computations in the appendix.
- In the paragraph "The existence and the uniqueness of Implicit-CKs and Implicit-NTKs", $z$ are not introduced before (I think they are only in the appendix), and it's confusing because another $z$ is used to define INNs in eq (1) and (2).
- the label vector $j$ could be called one-hot-encoded label vector this would help readers understand what it is more than the dirac notation I think.
- remark 1 is proved in the appendix, so it should be mentioned.


Refs:
[A] Micaelli, Paul, et al. "Recurrence without Recurrence: Stable Video Landmark Detection with Deep Equilibrium Models." Proceedings of the IEEE/CVF Conference on Computer Vision and Pattern Recognition. 2023.
[B] Fung, Samy Wu, et al. "Jfb: Jacobian-free backpropagation for implicit networks." Proceedings of the AAAI Conference on Artificial Intelligence. Vol. 36. No. 6. 2022.
[C] Bai, Shaojie, et al. "Deep equilibrium optical flow estimation." Proceedings of the IEEE/CVF Conference on Computer Vision and Pattern Recognition. 2022.
[D] Ramzi, Zaccharie, et al. "SHINE: SHaring the INverse Estimate from the forward pass for bi-level optimization and implicit models." arXiv preprint arXiv:2106.00553 (2021).
[E] Bai, Shaojie, Vladlen Koltun, and J. Zico Kolter. "Neural deep equilibrium solvers." International Conference on Learning Representations. 2021.

**Questions:**

- why are some assumptions named condition?
- why is it important to study CKs to show equivalence between INNs and ENNs?
- is it possible to have a bias in the parametrization of INNs in this work?
- how relevant is assumption 3?

**Details Of Ethics Concerns:**

This is somewhat minor in the sense that it's the only part I checked, but I noticed that the first paragraph of the introduction as well the first part of the second paragraph of the introduction are copy-pasted with a few language changes from "Global Convergence of Over-parameterized Deep Equilibrium Models" Ling et al. 2023 (I happen to be reading it for another review).
Of course it's only the introduction and it's not like it's a major part of it, but it raises concerns as to other parts of the submission, especially since this work is not even cited...

---

> ### Author Response · Authors · 2023-11-16
>
> We thank the reviewer for his/her careful reading and valuable feedback.
> In the following, we provide a step-by-step response to all comments raised by the reviewer.
> We hope in light of our responses the reviewer will consider raising the score accordingly.
>
> 1. **Code**: We have uploaded the code; please see the `INN\_Equiv\_ENN` folder in the supplementary material.
>
> 2.  **Interest of CK and NTK**:
> The CK and NTK have been studied extensively in the literature to assess the convergence and generalization properties of deep neural networks models.
> In short, NTK is a specific kernel defined in the context of (deep) neural networks.
> During (gradient descent) training, the network parameters change and the NTK also evolves over time.
> It has been shown by (Jacot et al. 2018) and follow-up work that for sufficiently wide neural networks trained on gradient descent with small learning rate, (i) the NTK is approximately constant after initialization and (ii) running gradient descent to update the network parameters is **equivalent** to kernel gradient descent with the NTK.
> This duality allows one to assess the **training dynamics, generalization, and predictions** of wide neural networks as closed-form expressions involving eigenvalues and eigenvectors of the NTK.
> As already mentioned in Section 1.1, the NTK has been studied for different types of DNN ranging from convolutional, graph, to recurrent and implicit NN, see (Feng and Kolter, 2020).
> We have also expanded the related work discussion on NTK in Section 1.1 on how it applies to assess DNN models.
>
> 3. **Bias in INNs**:
> We thank the reviewer for pointing this out. This is indeed a limitation of the present analysis in INN.
> For the moment the proposed theoretical framework is *not* able to cover deterministic and/or random bias.
> To the best of our knowledge, the only work on precise high-dimensional asymptotics of DNN models that has taken the bias into account is [C1], but only on a single-hidden-layer explicit neural network model.
> It would be of future interest to extend the proposed analysis framework to cover deterministic or random bias, which may lead to further improvement on the network performance.
>
> 4. **Assumption 3**:
> The assumption of $n,p \to \infty$ with $p/n \to c \in (0,\infty)$ is imposed so that the present analysis is performed in a (mathematically rigorously) high-dimensional asymptotic setting.
> The conducted analysis here does **not** demand that $n,p$ be growing variables but merely that they be *both* large (for the large $n,p$ asymptotics to be accurate, say).
> Precisely, our derivations and results use this technically convenient means to approximately quantify the INN performance achieved for all large $n,p$ pair.
> As such, one may freely assume $p$ large but fixed and $n$ growing, without altering the conclusions of our present study (the approximation error for our results would then be of the order $O(p^{-1/2})$ instead of $O(n^{-1/2})$ in Theorem 1 and 2).
> This is in complete adequacy with modern machine learning practice where the data dimensions are generally in hundreds or even thousands.
> The GMM scaling (by $\sqrt p$) and the growth rate in Assumption 3 are commonly used in the literature of high-dimensional ML for a wide class of models ranging from kNN, LDA, SVM, and shallow neural network models, see [C2], Section 2 of [C3], and Section 4.1 of [C4].
> We have added the current **Remark 1** in the revised version for further discussion on this point.
>
> 5. **Explicit NNs**:
> In Section 4, we have compared in Figure 3 the performance of INNs, L-ReLU-ENNs and ReLU ENNs with the **same** width $m$.
> We have added the sentence "Implicit and explicit NNs share the *same* network width $m \in { 32, 64, 128, 256, 512, 1024, 2048, 4096, 8192 }$" to further clarify this setup.
> Moreover, as suggested by the reviewer, we compared the time costs of the inference and training of INNs and ENNs.
> The inference time cost of an INN is about $25\times$ that of an ENN with the same dimension.
> This is due to the fact that it takes around 25 iterations for an INN to reach the iteration error threshold (1e-3).
> Additionally, we observe that ENNs have remarkable advantage over INNs in terms of the training speed.
> We report a brief result in the following table (each NN is trained for $150$ epochs).
>
> |        | $m$         | 256     | 1024    | 8192 |
> |--------|-------------|---------|---------|------|
> |CIFAR-10| ENN         | 0.096h    | 0.097h    | 0.111h |
> |        | INN         | 1.667h | 2.21h | 11h  |
> |MNIST   | ENN         | 0.089h    | 0.095h   | 0.097h |
> |        | INN         | 1.333h | 2h      | 10h  |
> |F-MNIST | ENN      |  0.089h    | 0.095h   | 0.097h |
> |        | INN         | 1.5h | 2h      | 10h  |
> |        |          |  |      |   |

---

> > ### Author Response · Authors · 2023-11-16
> >
> > 6 **Relevance of the experiments**:
> >
> > 6.1 **approximation error as the dimensions grow**:
> > As suggested by the reviewer, we conduct experiments to demonstrate the approximation error as the dimension $p$ grows with $n$.
> > The experimental setting is the same as Figure 1, except that $p$ grows from $125$ to $2000$, and the corresponding $n$ increases from $100$ to $1600$, with $p/n=1.25$.
> > The results are reported in the following table, demonstrating that the approximation error decreases with the growth of $p$ and $n$.
> >
> > |  | |  | | | |
> > |-------------------|------|------|------|------|------|
> > | $p$               | 125  | 250  | 500  | 1000 | 2000 |
> > | $n$               | 100  | 200  | 400  | 800  | 1600 |
> > | $\|\|G^*-\bar{G}\|\|$ | 1.63 | 0.48 | 0.23 | 0.12 | 0.08 |
> > |  | |  | | | |
> >
> > 6.2 **about eigenvalues**:
> > The closeness between the corresponding eigenvalues and/or eigenvectors of $G^*$ and $\bar{G}$ follows from the spectral norm bound/convergence in Theorem 1, and standard Weyl's inequality and/or Davis-Kahan theorem.
> > We have chosen to plot the eigenvalues since (i) the eigenvalues and eigenvectors of (CK and NTK) are directly involved in the training dynamics, generalization, and predictions of wide DNN models (via the NTK) as discussed in Section 1.1; and (ii) the eigenvalues visualize better than eigenvectors.
> > We apology for any possible confusion.
> > For the right panels of Figure 1, we agree with the reviewer that the two eigenvalue plots (in red and blue) are **not** exactly the same, and they indeed should **not** be.
> > Since our theoretical results are **only** proven for Gaussian mixture data and MNIST data have no reason to be close to GMM in any sense, we should **not**, a priori, expect that our theoretical result hold on MNIST.
> > In practice, however, we **do** observe similar behavior between $G^*$ and $\bar{G}$ in the right panels of Figure 1 (for most eigenvalue bars as well as the few largest eigenvalues).
> > This, together with the NTK theory, can be directly used to derive the similar training dynamics and generalization performance of a given INN and a carefully designed explicit network, as numerically demonstrated in Figure 3 (a)-(c).
> >
> > 6.3 **performance on CIFAR-10**:
> >
> > We would like to stress that the aim of this paper is to explore the connection between implicit and explicit NNs, not to achieve SOTA performance on, say DEQs.
> >
> > To address the concern raised by the reviewer, we conduct additional experiments on CIFAR-10 with carefully pre-training and early stopping. This accuracy is observed to boost to nearly 0.85.
> > Please see **Figure 5** in Section F in the revised supplementary material.
> > The main modification from Figure 3-(c) involves using features from a pre-trained VGG model on CIFAR-10.
> > We then use the features generated by this VGG pre-trained model as the input for our network.
> > This is in contrast to the setting of Figure 3-(c) that were obtained using a VGG model pre-trained on ImageNet, leading to sub-optimal performance on CIFAR-10.
> >
> > - **Minor**:
> >
> > 1. We have polished the paper and fixed typos in the revised version.
> >
> > 2. We have cited the relevant literature suggested by the reviewer, please see the related work section in Section 1.1 of the revised paper.
> >
> > 3. As your suggested, we have revised the text and stated that it is a result of (Feng and Kolter, 2020, Theorem 2).
> >
> > 4. We thank the reviewer for this kind suggestion, but we decide to leave the (important) result on NTK in the main text, the study of which is now better motivated. For better readability, we have moved the proof of Corollary 1 in the appendix.
> >
> > 5. The $\mathbf{z_i}$ used in the paragraph "The existence and the uniqueness of Implicit-CKs and Implicit-NTKs" is defined in Eq. (1). We have clarified this point in the revised paper.
> >
> > 6. We thank the reviewer for this excellent comment. We would like to kindly point out that the vectors $\mathbf{j}_a$ are of dimension $n$ and are particularly *not* one-hot-encoded label vectors.
> > Nonetheless, the rows of $\mathbf{J} \in \mathbb{R}^{n \times K}$ are indeed standard one-hot-encoded label vectors, and we have added this clarification in the revised versions.
> >
> > 7. In the revised paper, we have mentioned that Remark 1 (the current Remark 2) is proven in **Lemma A.1** in the supplementary material as suggested.

---

> > > ### Author Response · Authors · 2023-11-16
> > >
> > > - **About text duplication with (Ling et al. 2023) in the first paragraph of the introduction**:
> > > We would like to assure the reviewer that this paper is **not** a case of plagiarism or dual submission.
> > > Our work exhibits distinct differences from (Ling et al. 2023).
> > > The focus of this paper is to explore the connection between implicit and explicit neural networks, while (Ling et al. 2023) focuses on the training dynamics of INNs.
> > > We kindly note that **only** the background introduction part are borrowed from (Ling et al. 2023) and the missing of the reference is not deliberate.
> > > To address your concern, we have revised the first paragraph of introduction and cited (Ling et al. 2023) in the related work section of the revised paper.
> > >
> > >
> > > [C1] Adlam, B., Levinson, J. A., & Pennington, J. (2022, May). A random matrix perspective on mixtures of nonlinearities in high dimensions. In International Conference on Artificial Intelligence and Statistics (pp. 3434-3457). PMLR.
> > >
> > > [C2] Couillet, R., Liao, Z., & Mai, X. (2018, September). Classification asymptotics in the random matrix regime. In 2018 26th European Signal Processing Conference (EUSIPCO) (pp. 1875-1879). IEEE.
> > >
> > > [C3] Blum, A., Hopcroft, J., & Kannan, R. (2020). Foundations of data science. Cambridge University Press.
> > >
> > > [C4] Couillet, R., & Liao, Z. (2022). Random matrix methods for machine learning. Cambridge University Press.

---

> ### Comment · Reviewer_9wKK · 2023-11-18
>
> I would like to thank the authors for engaging in the rebuttal process.
> I will answer point by point subsequently:
>
> - **Code**: the code has indeed been provided now, but it's not runnable as is (with the default instructions, for mnist it runs): I get the following error `FileNotFoundError: [Errno 2] No such file or directory: './pretrained/cifar10_vgg.pth'`. It seems that the pretrained directory has not been included, and probably shouldn't be because of its size. It would be better to give a way to obtain the weights from the internet (anonymously). Moreover, I would suggest removing all potential sources of de-anonymization like comments in the original language.
> Furthermore, I think it's important to follow the NeurIPS guidelines established [here](https://nips.cc/Conferences/2020/PaperInformation/CodeSubmissionPolicy), in particular point 7: for now we don't know what the dependencies are, and we don't know how to reproduce the figures because the code is not present.
> - **Questions about the code**: 1) It seems that the 2 classes `Explicit` (which is used for the leaky relu experiment) and `Explicit_relu` (used for the relu experiment) are the same. I therefore think that the difference between the 2 that is noticed in the paper might only be due to a different seed, indeed no seeding mechanism is used in `w_matrix = torch.randn(input_dimension, dim).to(args.device)`. It is very important to address this point because it is one of the main arguments of the paper that the ReLU network is less performant. 2) The experiments are not run over multiple seeds. It is therefore difficult to know if the observed gap is significant or not. 3) I am not sure I understand the implementation of the explicit NNs: the inner weight matrix is fixed and not learned. Why this choice? How does it correspond to the paper?
> - **Construction of the explicit NN**: It became apparent to me after reading the code that actually this part of how to exactly construct the explicit NN was not clearly laid out in the paper. How are the weights set? The paper just mentions the activations part.
> - **Bias**: I apologize: apparently Feng and Kolter also do not use a bias. Still I think it is important to comment on why this is problematic in the proof. I also think it's important to mention it clearly in the paper.
> - **Assumption 3**:
> > As such, one may freely assume $p$ large but fixed and $n$ growing, without altering the conclusions of our present study (the approximation error for our results would then be of the order $O(p^{-\frac12})$ instead of $O(n^{-\frac12})$ in Theorem 1 and 2). This is in complete adequacy with modern machine learning practice where the data dimensions are generally in hundreds or even thousands
>
> I think this is a misleading proposition. $O(p^{-\frac12})$ only makes sense when $p \to \infty$, otherwise it's just a constant and does not give the intended result which is that the approximation error should be small when the number of data points increases. In modern machine learning practice the data dimension is generally fixed (although it may be huge).
> I think however, that as other works (for example Mei and Montanari's), it would be useful to comment on the rate of the 2 limits.
> It would be interesting also to comment on why it is needed intuitively to use high dimensional asymptotics.
> - **Experiments**: I think it's nice to see this result: how can I reproduce it with the code? Further as I mentioned in my original review, could it be possible to do the same with real data? To me this result should replace Figure 1. which does not show anything quantitative and does not illustrate Theorem 1.
> - **Text duplication**: I do not think this paper is a severe case of plagiarism, as indeed I only saw the introduction being copy-pasted. However, I would say that even in the current revised version, it's problematic that the introduction is so similar to Ling et al. : it's now not copy-pasted but slightly reworked.

---

> ### Author Response · Authors · 2023-11-19
>
> We thank the reviewer for his/her reply.
> We are glad that some of the reviewer's concerns have been successfully addressed.
> We would like to further emphasize that our major contribution is:
> 1. to build a mathematically rigorous and **explicit** connection between explicit and implicit network model by focusing their CK and NTKs (that depends on the network design, activation, and data statistics, all in a rather explicit fashion); and
> 2. to provide empirical evidences to support the theoretical insight obtained from the aforementioned theory.
> This explicit theory comes (of course!) at the cost of some technical assumptions, for example the high-dimensional GMM setting, the NTK (that is computed from random weights), etc.
> Given that said, we  believe it is of interest and important to present this result to the INN community, which, to the best of knowledge, is the first **explicit** result on the equivalence between explicit and implicit NN to the community.
>
> The remaining comments mainly involved the definition and implementation of CK and NTK, the high-dimensional asymptotic setting considered in this paper, as well as some details to reproduce the figures.
> They are addressed respectively as follows.
>
> **Clarifications on CK, NTK and their implementations**:
> We would like to further clarify that, per their definition in Eqn. (4)-(6), the CK and NTK matrices are defined and computed for network having random weights, for both implicit and explicit networks, see Assumption 1 and the sentence "As in Assumption 1, we assume that weights matrices Ws have i.i.d. entries ..." after Eqn. (20).
> This is in perfect accordance with the line of works on NTK, see (Jacot et al., 2018) and (Feng and Kolter, 2020), in which the NTK has been shown to be approximately constant (over time) after random initialization for wide networks trained on gradient descent with small learning rate, see also the related work discussion in Section 1.1 of the revised version.
>
> In our experiments, we compare the eigenspectral behaviors of CK/NTK matrices with their approximations and/or with their explicit counterparts.
> By definition (and as stated **explicitly** in the caption), these matrices are computed for network having random weights (with expectation numerically estimated from sample means).
> For Figure 3, we have added a clarification sentence that "only the final readout layer of both implicit and explicit networks are trained, with all intermediate layer weights fixed at random, as in line with the NTK literature."
> We are running additional experiments that train *all* weights (instead of only the last readout layer) of the network.
> Note that such additional experiments is of (empirical) interest only to networks of small width $m$ since, for wide networks, the NTK convergence results established in (Jacot et al., 2018) and (Feng and Kolter, 2020) ensure the closeness of the performance between networks with random or trained inner weights matrices.
>
>
> **Assumption 3**: We thank the reviewer of his/her further comment. We apology if our previous reply has introduced further confusion to the reviewer, below are some further clarifications:
> 1. the present work is in the same vein as (Mei and Montanari, 2021) in the sense that here, in the high-dimensional asymptotics setting $n,p \to \infty$, the two dimensions retain a constant ratio $p/n \to c \in (0,\infty)$ as in (Mei and Montanari, 2021);
> 2. for the reviewer to have a better grasp of the implications of our theoretical result, we would like to argue that the number of data points **cannot** go to infinity either.
> In fact, in ML practice and for a given problem, the number of data points itself **cannot** increase (or at least, cannot increase to go beyond a certain prefixed number).
> And the approximation errors given in this paper (as well as in, e.g., Mei and Montanari, 2021) should be understood a (concentration-type) bounds saying that, with high probability, the error is bounded by some expression that is a **decreasing** function of both $n$ and $p$.
> A detailed description on how such bound explicitly depends on $n,p$ is beyond the scope of this paper.
> Here, we only proved that in the high-dimensional limit as $n,p \to \infty$ with $p/n \to c \in (0,\infty)$ as in (Mei and Montanari, 2021), the approximation error vanishes at a rate of $O(p^{-1/2})$ or $O(n^{-1/2})$.

---

> > ### Author Response · Authors · 2023-11-19
> >
> > **About code and experiments**:
> >  * We have uploaded the pretrained VGG model `./pretrained/cifar10_vgg.pth` and removed comments. We have added `fig3_cifar10.sh`, `fig3_fashion_mnist.sh` and `fig3_mnist.sh` for readers to reproduce Figure 3. Moreover, we have uploaded `readme.txt` and `requirements.txt` for specification of dependencies.
> >
> >  * We apologize for mistakenly copying  the 'Explicit' function to the `Explicit_relu` during the code organization process. `Explicit_relu` is implemented for a vanilla single-layer fully-connected ReLU NN. In the updated version code, we have fixed this issue. Moreover, we think that there is a misunderstanding regardig your statement that "the ReLU networks is less performant is one of the main
> > arguments of this paper." In fact, the main argument of our paper is that high-dimensional ENNs is equivalent to INNs through the use of carefully designed activation.  We only argue that the performance of the proposed L-ReLU-NN is more closely aligned with INN than that of arbitrarily designed ENN, e.g. ReLU ENN. This misunderstanding may have been caused by  our previous code. In the newly uploaded code, we have resolved this issue.
> >
> >  * We would like to assure you that the phenomenon observed in Figure 3 is not due to different random seeds. We are starting to run  the experiments over multiple-seeds. It may take a significant amount of time due to the training speed of INNs. Perhaps running a quick test on our newly uploaded code can partially alleviate your concerns.
> >
> >
> >  * About construction of the explicit NN: In our original paper, we have defined the exact form of the explicit NN in Eq. (18). Furthermore, just below Eq.(20), we define the distribution for the weights in the explicit NN.
> >
> >  * About bias: We thank the reviewer for the helpful comment. For detailed discussion on this point, please see Remark 4 in the revised supplementary material.
> >
> >  * About Figure 1: we have uploaded `error_wrt_p_n.m` to reproduce  the additional result of the approximation error as the dimension $p$ grows with $n$.

---

### Official Review · Reviewer_9F3o · 2023-10-31

**Soundness:** 2 fair
**Presentation:** 3 good
**Contribution:** 2 fair
**Rating:** 3
**Confidence:** 4

**Summary:**

This paper studies the connection between implicit and explicit neural networks. The authors show that, with well-designed activation functions and weight variances, a relatively shallow neural network is equivalent to the implicit neural network in terms of eigenspectra of the neural tangent kernel and conjugate kernels. Additionally, the authors complement their theoretical results with numerical experiments.

**Strengths:**

The paper studies an important question, that is, whether implicit neural networks have advantages over explicit neural networks. The authors is well written and easy to follow and without so many errors. The steup and results are clear.

**Weaknesses:**

1. A primary concern is the paper's ambitious claim, that is, **any** implicit neural networks is equivalent to relatively shallow networks in terms of the eigenspectra of NTK and CK. While the paper establishes this equivalence in Theorems 1 and 2, it does so under highly specific conditions. These conditions involve setting the variance hyperparameters to exceptionally small values, leading to forward propagation acting as a contraction mapping and converging at an exponential rate. This configuration may restrict the expressive capacity of implicit neural networks, effectively making them resemble relatively shallow networks. However, this setup is common. Consequently, it raises doubts about the validity of drawing broad conclusions based on this specific scenario. For instance, Neural ODEs, which are also categorized as implicit neural networks, do not adhere to Condition 1.

2. Another concern arises from the authors' requirement that the input data $x$ follow Gaussian mixtures. This approach might be restrictive since, for large p, the mean $\mu/\sqrt{p}$ and covariance $C/p$ tend to zero, resulting in inputs that are all close to zero. Under these conditions, Theorems 1 and 2 hold as the authors also assume both n and p approach infinity at the same rate. In such cases, the network may struggle to distinguish inputs $x$, making it less relevant to assess whether a network is shallow or deep. I'm not an expert in Gaussian mixture models, so please feel free to correct me if my understanding is incorrect. Additionally, I plan to review comments from other experts in this area if available.
3. The paper defines intermediate transitions in equations 1 and 2 but doesn't provide a clear definition of the neural network itself. This omission is notable because it's essential to have a precise understanding of the network's structure. Additionally, it's worth noting that $z^*$ cannot be considered the network's output, as its dimension varies depending on the network's design.
4. There is a typo in Equation (10) concerning the definition of T.
5. While the paper extensively studies NTK and CKs, it would be beneficial to complement the theoretical analysis with experiments that demonstrate the behavior of $|G^*-\Sigma^2|$ for *finite-width* networks using simulations. This would provide more practical insights into the implications of the findings.

**Questions:**

See the weakness.

---

> ### Author Response · Authors · 2023-11-16
>
> We thank the reviewer for his/her careful reading and valuable feedback that helps us improve the paper.
> In the following, we provide a step-by-step response to all comments raised by the reviewer.
> We hope in light of our responses the reviewer will consider raising his/her score accordingly.
>
> 1. **ambitious claim**: We definitely agree with the reviewer that our proposed analysis is specific to implicit neural networks built upon contractive mapping, e.g., DEQs, and the term "any implicit neural network" can indeed be misleading.
> In the revised version, we have added the sentence "by focusing on a typical implicit NN, the deep equilibrium model (DEQ)" in the introduction to clarify. And we have replaced "any implicit neural network" with **a given DEQ model** in the paper to make our statements more precise.
> We would also like to point out, as consolidated by the reviewer, that our Condition 1 on the variance parameter is commonly used and is consistent with previous studies on over-parameterized DEQs, see [B1-B3].
>
> 2. **requirement on the input data following Gaussian mixtures**:
> We would like to kindly point out that the Gaussian mixture model (GMM) in Equation (7) is nothing but standard multivariate Gaussian distribution $\mathcal{N}(\mu,C)$ in $\mathbb{R}^p$ normalized by $\sqrt{p}$.
> This is to ensure that the data vectors are of bounded Euclidean norm (with high probability), and is in line with previous studies on over-parameterized explicit or implicit NNs, see [B1-B2] and [B4-B5].
> Indeed, the GMM in Equation (7) and Assumption 3 have been studied in the literature of high-dimensional statistics and have been shown solvable for a wide class of machine learning (ML) models ranging from kNN, LDA, SVM, and shallow neural network models, see, e.g., [B6], Section 2 of [B7], and Section 4.1 of [B8].
> In particular, it is known (see [B6] and Section 2 of [B7]) that different ML methods with different hyper-parameters can lead to different classification performance under Assumption 3.
> Such theoretical results, as also in the case of our paper, can be exploited to the design of ML models and/or hyper-parameter tuning.
> We have added the current **Remark 1** to provide further discussion on this point.
>
>
> 3. **transitions in Eqs. (1) and (2) and network structure**:
> We thank the reviewer for this valuable suggestion.
> In the revised paper, we have clarified, after Eq. (2), that the network's output is given by $f(\mathbf{x})=\mathbf{a}^\top\mathbf{z^*}$ with readout vector $\mathbf{a}\in \mathbb{R}^m$.
> This, together with Eq.(1) and (2), provides a complete definition of a fully-connected DEQ model.
>
> 4. We have fixed typos in the revised paper.
>
> 5. **simulations on finite-width networks**:
> In Section D of the supplementary material of the revised paper, we provide in Table 1 additional experiments on the spectral norm difference $\|G^*-\Sigma^{(2)}\|$ for finite-width INNs.
> The experiment is conducted on GMM data under the same setting as Figure 4 and is repeated follows.
>
> |                        |       |       |      |      |    |      |      |  |    |
> |------------------------|-------|-------|------|------|------|------|------|------|------|
> | $m$                    | 32    | 64    | 128  | 256  | 512  | 1024 | 2048 | 4096 | 8192 |
> | $\|G^*-\Sigma^{(2)}\|$ | 17.32 | 10.24 | 6.53 | 3.71 | 1.60 | 0.93 | 0.81 | 0.12 | 0.12 |
> |                        |       |       |      |      |    |      |      |  |    |
>
> We particularly observe that (i) the difference $\|G^*-\Sigma^{(2)}\|$ for finite-width ploy-ENN decreases with the increase of the network width $m$, and (ii) the approximation saturates at a low level ($\sim 0.12$) and this is due to the finite $n, p$ as opposed to our asymptotic theoretical results in Theorem 1 and 2.

---

> > ### Author Response · Authors · 2023-11-16
> > **Reference**
> >
> > [B1] Ling, Z., Xie, X., Wang, Q., Zhang, Z., & Lin, Z. (2023, April). Global convergence of over-parameterized deep equilibrium models. In International Conference on Artificial Intelligence and Statistics (pp. 767-787). PMLR.
> >
> > [B2] Truong, L. V. (2023). Global Convergence Rate of Deep Equilibrium Models with General Activations. arXiv preprint arXiv:2302.05797.
> >
> > [B3] Gao, T., Liu, H., Liu, J., Rajan, H., & Gao, H. (2021, October). A global convergence theory for deep ReLU implicit networks via over-parameterization. In International Conference on Learning Representations.
> >
> > [B4] Mei, S., & Montanari, A. (2022). The generalization error of random features regression: Precise asymptotics and the double descent curve. Communications on Pure and Applied Mathematics, 75(4), 667-766.
> >
> > [B5] Bubeck, S., & Sellke, M. (2021). A universal law of robustness via isoperimetry. Advances in Neural Information Processing Systems, 34, 28811-28822.
> >
> > [B6] Couillet, R., Liao, Z., & Mai, X. (2018, September). Classification asymptotics in the random matrix regime. In 2018 26th European Signal Processing Conference (EUSIPCO) (pp. 1875-1879). IEEE.
> >
> > [B7] Blum, A., Hopcroft, J., & Kannan, R. (2020). Foundations of data science. Cambridge University Press.
> >
> > [B8] Couillet, R., & Liao, Z. (2022). Random matrix methods for machine learning. Cambridge University Press.

---

> > > ### Author Response · Authors · 2023-11-20
> > >
> > > We express our gratitude to the reviewers once more for their valuable comments. Kindly note that we have addressed all your feedback. We are ready and willing to address any additional concerns you may have.

---

> > > > ### Comment · Reviewer_9F3o · 2023-11-21
> > > >
> > > > I appreciate the authors' response to my concerns; however, they remain unresolved. After reviewing other reviewers' comments, I've encountered further concerns.
> > > >
> > > > Specifically, reviewer 9wKK raises questions about CK and NTK, where CK refers to the NNGP kernel for neural networks under the establishment of NNGP correspondence. In this context, it pertains to the NNGP kernel for DEQ. Notably, the neural network DEQ is **not initially defined** in the authors' first submission but is now clarified after equation (2) in their revision. This NNGP kernel or CK aims to characterize the limiting covariance of $f(x_i)$ and $f(x_j)$ as width $m \rightarrow \infty$. Essentially, it represents the limit of $\langle z_i^*, z_j^* \rangle$ as $m \rightarrow \infty$. However, it's important to note that this limit is **not** $G_{ij}^*$ as introduced in [1], where DEQs are not officially defined either; rather, it should be $G_{ij}^* - \sigma_b^2 \langle x_i, x_j \rangle$, a precise derivation presented in [2]. Unfortunately, the subsequent analyses and results heavily rely on this foundational result, and a notable **flaw** has been identified. Given this, I've chosen to maintain my score at this moment.
> > > >
> > > > [1] Feng, Zhili, and J. Zico Kolter. "On the neural tangent kernel of equilibrium models." arXiv preprint arXiv:2310.14062 (2023).
> > > >
> > > > [2] Tianxiang Gao, Xiaokai Huo, Hailiang Liu, Hongyang Gao. "Wide Neural Networks as Gaussian Processes: Lessons from Deep Equilibrium Models." NeurIPS 2023.

---

### Official Review · Reviewer_aCT6 · 2023-11-03

**Soundness:** 3 good
**Presentation:** 3 good
**Contribution:** 3 good
**Rating:** 5
**Confidence:** 3

**Summary:**

The authors show that the conjugate kernel (CK) and the neural tangent kernel (NTK) of implicit NNs can be approximated by surrogate kernels, based on an operator norm, when the data are distributed according to a particular Gaussian mixture model. In addition, they show how to construct shallow explicit NNs for which the associated CK and NTK are close to the surrogate kernels of the implicit NNs. The claim is that any implicit NN can be approximated by a shallow explicit NN. In the experiments, they demonstrate the performance of the proposed explicit NNs compared to the original implicit NNs.

**Strengths:**

- The idea of constructing efficient explicit NNs to approximate implicit NNs is very interesting.
- The technical part of the paper seems solid and sensible, however, I have not verified the theoretical results.
- In the experiments, the proposed approximate explicit NNs seem to work as expected similar to the original implicit NNs.

**Weaknesses:**

- The paper is fairly well written for a mainly theoretical work. However, I think that the text can be improved to become more accessible to non-experts in the implicit NNs field. For example, the proof of Corollary 1 can be moved to appendix to save space.
- The theoretical results rely on many assumptions, for example, the distribution of the data to follow this particular Gaussian mixture model. When $p$ is high, it seems that the associated GMM becomes degenerate.

**Questions:**

The surrogate kernels $\overline{G}$ and $\overline{K}$ are supposed to be random matrices, but in which sense, as they seem to be expectations.

---

> ### Author Response · Authors · 2023-11-16
>
> We thank the reviewer for his/her careful reading and valuable feedback.
> In the following, we provide a step-by-step response to all comments raised by the reviewer.
>
> 1. **text can be improved to become more accessible**:
> We thank the reviewer for this helpful suggestion and we have moved the proof of Corollary 1 to Section D in the supplementary material and provide only a proof sketch in the main text of the revised paper.
>
> 2. **about the assumption on GMM data**:
> We would like to kindly point out that the Gaussian mixture model under study in Equation (7) and Assumption 3 is *not* degenerate for $p$ large.
> Instead, it is nothing but standard multivariate Gaussian distribution in $\mathbb{R}^p$ normalized by $\sqrt{p}$.
> This normalization is in line with the previous studies on over-parameterized explicit or implicit NNs, see [A1-A2] and [A3-A5], and more generally with the literature of high-dimensional statistics and random matrix theory.
> This is necessary to ensure that (with high probability) the Euclidean norm $\| \|x_i\|\|$ of data vectors remains bounded in the high-dimensional limit as $n,p \to \infty$ (also for large but finite $p$, see Section 2 of [A6] and Section 4.1 of [A7]).
> Under the normalization in Equation (7),  the growth rate in Assumption 3 is necessary and can in fact be shown to be optimal in a Neyman–Pearson sense (e.g., Gaussian mixture with closer means is indistinguishable for large $p$),  see [A8].
> We have added the current **Remark 1** to further elaborate on this point.
>
> 3. **randomness in $\bar{G}$ and $\bar{K}$**:
> The expectations in the definition of CK $G$ and NTK $K$ are taken with respect to the random weights, and the randomness from the input GMM data still exists.
> Consequently, $G$ and $K$, as well as their approximations $\bar{G}$ and $\bar{K}$,  are still random matrices for random input GMM data.
> We have added the current **Footnote 1** to clarify.
>
>
> [A1] Mei, S., & Montanari, A. (2022). The generalization error of random features regression: Precise asymptotics and the double descent curve. Communications on Pure and Applied Mathematics, 75(4), 667-766.
>
> [A2] Bubeck, S., & Sellke, M. (2021). A universal law of robustness via isoperimetry. Advances in Neural Information Processing Systems, 34, 28811-28822.
>
> [A3] Ling, Z., Xie, X., Wang, Q., Zhang, Z., & Lin, Z. (2023, April). Global convergence of over-parameterized deep equilibrium models. In International Conference on Artificial Intelligence and Statistics (pp. 767-787). PMLR.
>
> [A4] Truong, L. V. (2023). Global Convergence Rate of Deep Equilibrium Models with General Activations. arXiv preprint arXiv:2302.05797.
>
> [A5] Feng, Z., & Kolter, J. Z. (2023). On the neural tangent kernel of equilibrium models. arXiv preprint arXiv:2310.14062.
>
> [A6] Blum, A., Hopcroft, J., & Kannan, R. (2020). Foundations of data science. Cambridge University Press.
>
> [A7] Couillet, R., & Liao, Z. (2022). Random matrix methods for machine learning. Cambridge University Press.
>
> [A8] Couillet, R., Liao, Z., & Mai, X. (2018, September). Classification asymptotics in the random matrix regime. In 2018 26th European Signal Processing Conference (EUSIPCO) (pp. 1875-1879). IEEE.

---

> > ### Comment · Reviewer_aCT6 · 2023-11-22
> > **Post-rebuttal**
> >
> > I would like to thank the authors for answering my questions and improving the manuscript based on the comments of the reviewers. However, after reading the rest of the reviews, there seem to be some technical flaws that cannot be addressed with a few updates in the current manuscript. Therefore, I will reduce my score to 5, as I still like the general idea but the paper is not ready for publication.

---

### Author Response · Authors · 2023-11-20
**Global response to Reviewers**

We would first like to thank the reviewers for their detailed remarks that help us improve this manuscript.
The comments of utmost importance involved the setting of high-dimensional GMM data, as well as the experiment details in Section 4 of our submission. In a nutshell,

1. The GMM under study is nothing but standard multivariate Gaussian  $\mathcal{N}(\mu,C)$ normalized by the square root dimension $\sqrt p$.
This "normalization" scaling is standard in the literature of high-dimensional statistics and its application to (e.g., over-parameterized) machine learning models.
In particular, the resulting GMM is **not** degenerate and has been studied previously in kNN, LDA, SVM, and shallow neural network models.
In the revised version of the paper, we have added Remark 1 to provide further discussions on this point.

2. We have provided additional experiments as reviewers suggested, e.g. the empirical spectral norm difference w.r.t $p$, $n$ and $m$, novel experiments on CIFAR-10 data. Moreover, we have uploaded the code for readers to reproduce our empirical results.


 A step-by-step response to all comments raised by the reviewers has been provided in the following, with revisions marked in blue color in an updated version of the manuscript.

We eagerly await further comments from the reviewers and are prepared to address them.
We hope that, in light of our responses, the reviewers will consider raising their scores.